# Adjuvant Pancreatic Cancer Management: Towards New Perspectives in 2021

**DOI:** 10.3390/cancers12123866

**Published:** 2020-12-21

**Authors:** Anthony Turpin, Mehdi el Amrani, Jean-Baptiste Bachet, Daniel Pietrasz, Lilian Schwarz, Pascal Hammel

**Affiliations:** 1UMR9020-UMR-S 1277 Canther-Cancer Heterogeneity, Plasticity and Resistance to Therapies, University of Lille, CNRS, Inserm, CHU Lille, Institut Pasteur de Lille, F-59000 Lille, France; anthony.turpin@chru-lille.fr; 2Medical Oncology Department, CHU Lille, University of Lille, F-59000 Lille, France; 3Department of Digestive Surgery and Transplantation, Lille University Hospital, F-59000 Lille, France; Mehdi.ELAMRANI@chu-lille.fr; 4Department of Hepatogastroenterology and GI Oncology, La Pitié-Salpêtrière Hospital, INSERM UMRS 1138, Université de Paris, F-75013 Paris, France; jean-baptiste.bachet@aphp.fr; 5Department of Digestive, Oncological, and Transplant Surgery, Paul Brousse Hospital, Paris-Saclay University, F-94800 Villejuif, France; daniel.pietrasz@wanadoo.fr; 6Department of Digestive Surgery, Rouen University Hospital and Université de Rouen Normandie, F-76100 Rouen, France; Lilian.Schwarz@chu-rouen.fr; 7Service d’Oncologie Digestive et Médicale, Hôpital Paul Brousse (AP-HP), 12 Avenue Paul Vaillant Couturier, F-94800 Villejuif, France

**Keywords:** pancreatic cancer, adjuvant therapy, neoadjuvant therapy, biomarkers, precision medicine, timing

## Abstract

**Simple Summary:**

In operated pancreatic cancer patients who are able to begin treatment within 3 months after surgery, adjuvant chemotherapy is currently used to limit disease recurrence but questions remain for the clinician. Recently, modified FOLFIRINOX has become the standard-of-care in the non-Asian population, nevertheless there is still a risk of toxicity and feasibility may be limited in heavily pre-treated patients. Gemcitabine-Nabpaclitaxel, Gemcitabine alone in non-Asian patients are alternatives to be discussed. In Asia, S1-based chemotherapy remains the standard. The aim of this review is to summarize adjuvant management of resected pancreatic cancer and to raise current and future concerns, especially the need for biomarkers and the best holistic care for patients.

**Abstract:**

Adjuvant chemotherapy is currently used in all patients with resected pancreatic cancer who are able to begin treatment within 3 months after surgery. Since the recent publication of the PRODIGE 24 trial results, modified FOLFIRINOX has become the standard-of-care in the non-Asian population with localized pancreatic adenocarcinoma following surgery. Nevertheless, there is still a risk of toxicity, and feasibility may be limited in heavily pre-treated patients. In more frail patients, gemcitabine-based chemotherapy remains a suitable option, for example gemcitabine or 5FU in monotherapy. In Asia, although S1-based chemotherapy is the standard of care it is not readily available outside Asia and data are lacking in non-Asiatic patients. In patients in whom resection is not initially possible, intensified schemes such as FOLFIRINOX or gemcitabine-nabpaclitaxel have been confirmed as options to enhance the response rate and resectability, promoting research in adjuvant therapy. In particular, should oncologists prescribe adjuvant treatment after a long sequence of chemotherapy +/– chemoradiotherapy and surgery? Should oncologists consider the response rate, the R0 resection rate alone, or the initial chemotherapy regimen? And finally, should they take into consideration the duration of the entire sequence, or the presence of limited toxicities of induction treatment? The aim of this review is to summarize adjuvant management of resected pancreatic cancer and to raise current and future concerns, especially the need for biomarkers and the best holistic care for patients.

## 1. Introduction

Based on current incidence rates pancreatic ductal adenocarcinoma (PDAC) is expected to become the second leading cause of death from cancer in the United States and Europe by 2030 [1,2]. It remains the gastro-intestinal cancer with the worst prognosis, with a five-year overall survival (OS) rate of 5–7% [3]. Most patients are diagnosed at advanced stages i.e., locally advanced (30%) or metastatic (>50%) and 60% to 80% patients who undergo curative-intent resection have tumor relapse within 3 years after surgery [4].

Adjuvant chemotherapy is recommended in all operated patients who are in acceptable condition after surgery, regardless of pTNM stage. Although certain guidelines recommend starting chemotherapy within 3 months after surgery [4,5,6,7] this may be difficult because recovery may take longer. It has also been suggested that the date of initiation of adjuvant chemotherapy may be less important than administering the entire treatment, i.e., six months [8]. 

Good clinical practice guidelines for adjuvant treatment in PDAC were recently updated after more effective chemotherapy regimens were tested between 2016 and 2019 in four phase III studies [9,10,11,12]. However, these guidelines probably will change in the near future, for two main reasons. First, because of the increasing number of patients receiving a neoadjuvant treatment before surgical resection and also the rapid progress being made in the understanding of tumor biology, which could help to guide the choice of adjuvant treatment. 

The aim of this paper is to present a state-of-the-art review of adjuvant treatment following surgical resection of PDAC and to identify future directions for research.

## 2. Summary of Past Experience in Adjuvant Treatment of Operated PDAC

### 2.1. Chemotherapy

Twenty-two years ago, for the first time the large phase III ESPAC-1 trial by Neoptolemos et al. showed a benefit of adjuvant chemotherapy using 5-fluorouracil (5-FU modulated by leucovorin) [13]. Although much has been written about this study, in particular in relation to the methodological aspects of radiation therapy, with inferior and even deleterious results, OS was significantly increased in the 5-FU arm and the first stone was laid for adjuvant treatment. The German group CONKO then showed the efficacy of gemcitabine for this purpose [14] while the ESPAC-3 trial did not show any difference in efficacy between 5-FU and gemcitabine [15]. Finally, the ESPAC-4 trial reported that the combination of gemcitabine plus capecitabine was superior to gemcitabine: median OS: 28.0 months (95% CI 23.5–31.5) versus 25.5 months (22.7–27.9) HR: 0.82 (95% CI 0.68–0.98), *p* = 0.032) [11] (Table 1).

In 2018, since the publication of the PRODIGE 24 trial, modified FOLFIRINOX (mFOLFIRINOX), which was found to be significantly more effective than the reference gemcitabine regimen, has become the international standard in non-Asian patients [9,16]. After a median follow-up of 33.6 months, the trial reached its main objective of increasing disease-free survival (DFS). Median DFS was 21.6 months in the mFOLFIRINOX group versus 12.8 months in the gemcitabine group (HR stratified for one cancer event, second cancer or death: 0.58; 95% CI 0.46–0.73; *p* < 0.001). In addition, the secondary objective of OS was significantly better in the mFOLFIRINOX arm, with a median of 54.4 months versus 35.0 months for gemcitabine (stratified HR for death: 0.64; 95% CI, 0.48–0.86; *p* = 0.003). The results for DFS at 3 years, or metastatic-free survival, were significant and varied in the same way in favor of mFOLFIRINOX. In this study the patient population was highly selected and in good condition (PS 0–1), with centralized review of surgical and pathology reports, a normal postoperative CT scan and low CA 19.9 serum levels (<180 U/mL) [17]. These strict criteria were not required in former trials such as ESPAC-4 and suggested that the PRODIGE 24 trial may have included patients in better condition with less early metastatic relapse [17]. Nevertheless, the superiority of mFOLFIRINOX over gemcitabine was confirmed, especially because this difference was obtained while the results in the gemcitabine arm (OS 30 months) were the best reported so far. In addition, the DFS of 12.8 months in the gemcitabine arm was similar to that observed in the CONKO-001, ESPAC-4 and other trials, which does not support a selection bias [17].

In 2019, the APACT trial did not find that DFS was better with the gemcitabine-nab-paclitaxel combination than with the gemcitabine reference regimen [10]. After a median follow-up of 38.5 months, the estimated DFS was 19.4 months in the GemNab arm by an independent review and 18.8 months in the reference arm (HR: 0.88; 95% CI 0.729–1.063; *p* = 0.1824). OS in the interim analysis was 40.5 months and 36.2 months in the GemNab and gemcitabine arms, respectively (HR: 0.82 [0.680–0.996], *p* = 0.045) [10]. Despite this, updated NCCN guidelines propose GemNab as an alternative to mFOLFIRINOX [5]. These adjuvant polychemotherapies should be limited to patients in acceptable condition and with satisfactory biological parameters. The full paper on the APACT trial is pending. The results of both this study and that of PRODIGE 24 will be updated with a longer follow-up.

An alternative in patients who are not eligible for mFOLFIRINOX is the GemCap combination of gemcitabine and capecitabine, which was tested in the European ESPAC4 trial. In that study, 60% of patients had positive margins and 80% had lymph node involvement, which was higher than in any other chemotherapy trial. OS was better with GemCap in the population with negative margins (median OS: 39.5 vs. 27.9 months; *p* < 0.001). Median DFS was similar in the GemCap and the gemcitabine arms (see Table 1) and there was no increase in treatment-related grade 3 and 4 adverse events by adding capecitabine (24% with the combination versus 26% with gemcitabine; *p* > 0.05) [11]. 

S1-based chemotherapy, an oral therapy containing tegafur (5-FU prodrug), potassium oteracil and gimeracil, has become a standard of care in Asia since the Japanese trial JASPAC-01 results were reported [12]. This non-inferiority Phase III trial compared S1 to standard gemcitabine for 6 cycles in the adjuvant setting. The primary endpoint of this trial was per-protocol OS. The S1 compound was found to be non-inferior with a HR of mortality: 0.57 (95% CI 0.44–0.72; *p* < 0.0001 for non-inferiority and *p* < 0.0001 for superiority). In addition, overall 5-year survival was 44.1% (95% CI 36.9–51.1) in the S1 group and 24.4% (95% CI 18.6–30.8) in the gemcitabine group. The safety of S1 was found to be acceptable.

In conclusion, mFOLFIRINOX should be the option of choice rather than gemcitabine in eligible non-Asian patients [9]. In more frail patients, a gemcitabine-based chemotherapy is still a suitable option [5]. Gemcitabine or 5FU as monotherapy is an option in France [4]. In Asian patients, S1 is the standard adjuvant treatment [18]. 

### 2.2. Radiation Therapy

It is impossible to reach a conclusion on the role of chemoradiotherapy (CRT) in the adjuvant setting based on previous studies comparing chemotherapy and CRT. There was no significant increase in OS with CRT compared to the control group in the large prospective trials ESPAC-1 or EORTC [19,20] and van Laethem study [21]. On the other hand, two population-based studies using a national cancer registry database reported that CRT was more effective than systemic chemotherapy (SCT) [22,23]. In the series by You et al. [24] 335 patients received CRT (*n* = 65), SCT (*n* = 62) or CRT plus systemic chemotherapy (SCT) (*n* = 208) in an adjuvant setting. Overall median OS was 33.3 mo (95% confidence interval (CI): 27.4–38.6). There was no difference in median OS in the CRT group in patients with stage I/II cancer, (27.0 months (95% CI: 2.06–89.6), (35.8 mo (95% CI: 26.9–NA) in the SCT group and the CRT plus SCT groups (38.6 mo (95% CI: 33.3–55.7). In contrast, in the group of 59 patients with stage III PDAC, median OS was longer in the SCT group [19.0 mo (95% CI: 12.6-NA)] and the CRT-SCT group [23.4 mo (95% CI: 22.0–44.4)] than in the CRT group [17.7 mo (95% CI: 6.8–NA); *p* = 0.011 and *p* < 0.001, respectively]. The rate of adverse events was higher in the SCT and CRT-SCT groups than in the CRT group. 

## 3. The Current Limitations of Available Therapeutic Options

### 3.1. GemCap: Methodological Limitations

The methodology of the ESPAC 4 study was criticized, in particular because of the absence of a post-operative CT scan at inclusion and, especially, for the lack of significant benefit in DFS (*p* = 0.082) with OS curve separation only beginning late after 2 years [25]. The inclusion of patients with a poor prognosis into the groups creates an imbalance (e.g., 11% venous resection in the GemCap group vs. 17% in the control group) and may have influenced the results in favor of the combined regimen. In addition, a high post-operative CA19-9 serum value, a major independent prognostic factor, was found in 17% of patients, suggesting the presence of early metastatic disease. This might explain the superiority of the GemCap combination over gemcitabine alone at a CA19-9 level of >92.5 IU/mL.

### 3.2. FOLFIRINOX: Not for Everyone

Up to 30% of patients do not receive adjuvant therapy [26]. Modified FOLFIRINOX cannot be administered if patients have not fully recovered from major surgery in the presence of fatigue, weight loss, denutrition or diarrhea. In the PRODIGE 24 study by Conroy et al. [9], grade 3–4 adverse events were reported in 75.9% and 52.9% of patients in the mFOLFIRINOX and gemcitabine groups, respectively. The use of irinotecan was significantly associated with grade 3 or 4 diarrhea (adjusted odds ratio, 6.0; 95% CI 2.9–12.8; *p* < 0.001). In addition, the completion rates of adjuvant chemotherapy were 66.4% and 79%, respectively. However, this rate is similar to that reported in other therapeutic trials (60–70%) [27].

### 3.3. APACT Study: Main Objective with GemNab Combination (DFS) Was not Achieved Despite OS Was Improved

The results of the APACT study have been a topic of debate, in particular about the relevance of the primary endpoint (DFS) based on a blind analysis, rather than the investigator. Indeed, the blind analysis was performed on the basis of imaging alone, with no access to clinical data. To characterize PDAC recurrence on CT imaging can be difficult due to anatomical changes and fibro-inflammatory features in the tumor bed [28]. Median DFS and interim OS with GemNab and gemcitabine, assessed by independent reviewers was 19.4 months vs. 18.8 months, respectively (HR, 0.88; 95% CI, 0.729–1.063; *p* = 0.1824) and 40.5 months vs. 36.2 months (HR, 0.82; 95% CI, 0.680–0.996; nominal *p* = 0.045). The figures were much higher in the gemcitabine arm than those reported in the previous trials alone such as CONKO-001, which were 22.8 months and 13.4 months, respectively. However, the HRs for DFS and OS were both superior to 0.8 in the APACT trial and despite the statistically significant OS, the clinical benefit was limited. 

### 3.4. Particularity of S1 Regimen

Because the S1 regimen is not available outside Asia this drug has not been extensively evaluated in non-Asian patients [12].

## 4. New Issues

Today, there are new avenues of research in the adjuvant setting. Patients with borderline or even locally advanced PDAC may be candidates for secondary surgical resection after receiving induction treatment with intensive chemotherapy alone or followed by CRT. These treatments are now also being tested in a neoadjuvant setting in patients with resectable PDAC (example of PRODIGE-PANACHE ongoing studies).

### 4.1. Induction Strategy

The term “induction” treatment rather than “neoadjuvant” treatment is suitable for borderline/locally advanced tumors. Once again, gemcitabine is still the international standard for locally advanced PDAC (LAPC) [5,7] based on past studies [29]. Intensified regimens such as FOLFIRINOX or GemNab are now recognized options [5,6] based on a meta-analysis of retrospective studies, while the results of the French prospective study NEOPANC (FOLFIRINOX versus gemcitabine in LAPC) are pending. For example, Janssen’s meta-analysis of 24 studies (8 prospective, 16 retrospective) with FOLFIRINOX published in 2019 reported a resection rate of 67.8% and a R0 resection rate of 83.9% in patients who were able to undergo surgery [30]. Concerning GemNab, the results of the GAP study, the only randomized study confronting GemNab vs. Gem in this setting, have shown positive results, especially in terms of distant relapses [31].

There have been no well-designed, prospective randomized trials as yet, and no robust metanalysis has shown a benefit to survival with induction strategies. The main weakness of published studies is the lack of statistical power and the heterogenous populations (for example, the pooling of resectable, borderline and locally advanced tumors). In addition, these studies are limited by the small populations, low-volume centers and lack of consensus on resectability criteria after induction chemotherapy [26,32]. International trials are ongoing and the first prospective results are being reported. The LAPACT phase II open-label trial evaluating induction chemotherapy with six cycles of nab-paclitaxel 125 mg/m^2^ plus gemcitabine 1000 mg/m^2^ (days 1, 8, and 15 of each 28-day cycle) in patients with LAPC showed good tolerability and a promising efficacy. Median time to treatment failure was 9.0 months (90% CI 7.3–10.1), progression-free survival (PFS) was 10.9 months (90% CI 9.3–11.6), and OS was 18.8 months (90% CI 15.0–24.0). During induction therapy, 83 patients achieved disease control [77.6% (90% CI 70.3–83.5)] but 17 (16%) underwent surgery in the experimental arm (seven had R0 resection status, nine had R1). Toxicity was manageable and mainly hematological (grade 3 neutropenia, anemia and fatigue). There was no adjuvant chemotherapy following the induction strategy [33].

The recent multicenter phase III PREOPANC study in patients with resectable or borderline PDAC did not report any significant benefit to OS (primary endpoint) with preoperative CRT with gemcitabine versus frontline surgery followed by adjuvant gemcitabine. The median overall survival by intention-to-treat-analysis was similar in both arms, 16.0 and 14.3 months with preoperative CRT and frontline therapy, respectively (hazard ratio, 0.78; 95% CI, 0.58 to 1.05; *p* = 0.096). Nevertheless, administration of CRT was associated with a better R0 resection rate, 71% (51/72) compared to 40% (37/92) (*p* < 0.001) and better DFS and locoregional failure-free intervals than frontline surgery [34]. The results of the prospective randomized phase 2 ESPAC-5F trial, which evaluated the value of neoadjuvant treatment in borderline PDAC, were presented at the virtual ASCO meeting in 2020 [35]. Four treatments were compared in this trial: frontline surgery, 2 months of GemCaP, 2 months of mFOLFIRINOX and CRT (50.4 Gy with capecitabine). There was no difference between the frontline and induction treatment (pooled) arms for the primary endpoint, R0-R1 resection rate (62% versus 55%) or R0 (15% versus 23%). The pN+ rate in operated patients was lower in those who received CRT (25%) compared to other treatments (GemCap: 58%; mFOLFIRINOX: 73%; surgery upfront: 90%). The one-year survival rate was 77% in the induction treatment arm and 40% in the frontline surgery arm (HR = 0.27; 95% CI: 0.13–0.55; *p* < 0.001). Interestingly, the rate of adjuvant chemotherapy administered was similar regardless of the induction treatment arm: frontline surgery arm: 53%, GemCap induction: 50%, mFOLFIRINOX: 45%, CRT: 44%. These data are consistent with those of the PREOPANC study with gemcitabine where this regimen was found to be beneficial to patients with borderline PDAC. Altogether, these results suggest the value of induction treatment in patients with borderline PDAC.

Other prospective trials are needed to evaluate perioperative strategies with induction chemotherapy in patients with R0-unresectable or resectable PDAC [27]. It has been suggested that the administration of CRT after chemotherapy could result in significant downsizing or downstaging with increased R0 resection and fewer post-operative complications and the phase II PANDAS-PRODIGE 44 study in France (NCT02676349 is evaluating this question [36,37,38]. Further studies in stereotactic radiotherapy are also needed for purpose [27].

### 4.2. Which Adjuvant Treatment in Operated Patients Who Receive Neoadjuvant or Induction Strategy?

The role and modality of adjuvant treatment (type, duration) in patients who have undergone surgical resection after induction treatment have not been defined. This will probably be affected by preoperative toxicity, in particular neurotoxicity and the pathological response in the resected specimen. Certain authors prefer to repeat the administration of induction chemotherapy postoperatively [25,27], an approach that may not be possible in patients with a poor pathological response, for example for with a CAP Score of 3. 

In the study by Pietrasz et al., the administration of adjuvant chemotherapy in 54% of the 80 operated patients (borderline or locally advanced PDAC) after a median of 6 cycles of neoadjuvant FOLFIRINOX did not improve survival (HR, 0.85; 95% CI, 0.45–1.61; *p* = 0.62) [39]. Van Roessel et al. evaluated the role of adjuvant therapy in 520 patients with resectable (48.4%), borderline (41.4%) or locally advanced (10.2%) PDAC who a received a median of 6 cycles of FOLFIRINOX prior to surgical resection. Overall, 66.0% of patients received adjuvant chemotherapy as follows: FOLFIRINOX in 19.8%, gemcitabine-based chemotherapy in 58.6%, capecitabine in 4.1%, a combination or other agents in 13.1%, or an unknown type of treatment in 4.4%. Median OS was 38 months (95% CI, 36–46 months) after diagnosis and 31 months (95% CI, 29–37 months) after surgery. No difference in survival was found in patients who received adjuvant chemotherapy compared to those who did not (median OS, 29 vs. 29 months, univariate hazard ratio [HR], 0.99; 95% CI, 0.77–1.28; *p* = 0.93). Adjuvant chemotherapy was only found to be beneficial on multivariate analysis, in patients with a positive lymph node status (median OS, 26 vs. 13 months; multivariate HR, 0.41 [95% CI, 0.22–0.75]; *p* = 0.004). It is important to note that there was no further advantage in patients who received ≥ 4 cycles of FOLFIRINOX induction [40]. It is not known whether patients had N0 status before induction treatment or if this was due to downstaging by chemotherapy. A report by Skau Rasmussen et al. in 623 patients also found that adjuvant therapy following frontline surgery for pancreatic cancer only improved survival in patients with ypN+ PDAC [41]. Another study by Perri et al. in 245 patients who received induction treatment before surgical resection, reported that adjuvant treatment was marginally associated with longer OS (HR, 0.55; 95% CI, 0.29–1.01; *p* = 0.05). A subgroup analysis was not available [42]. Whether adjuvant treatment would be more suitable in patients with initial borderline or surgically treated, locally advanced PDAC rather than in those with frontline resectable tumors is an interesting question [41].

#### 4.2.1. Adjuvant Treatment according to Response to Induction Treatment

##### Lymph Node Ratio

The lymph node ratio (LNR), defined as the number of lymph nodes with metastatic disease among the total number of retrieved lymph nodes, has been validated as a prognostic factor in patients with PDAC [43]. In a retrospective study by Roland et al., the administration of adjuvant chemotherapy following induction treatment was associated with improved OS and time-to-recurrence in patients with low lymph node involvement (LNR < 0.15). Interestingly, patients with a significant lymph node burden following neoadjuvant treatment did not benefit from adjuvant chemotherapy in that study [43]. In contrast, in three large retrospective studies, the benefit of adjuvant therapy in patients who had received neoadjuvant therapy was limited to those with pathological node-positive status [40,41,44]. Thus, prospective studies are needed to evaluate the role of adjuvant chemotherapy, particularly in pN0 patients.

##### R0 Resection Rate

The R0 resection rate is also an important prognostic factor in operated PDAC [45]. A recent meta-analysis of 27 studies has suggested that survival following surgery after induction treatment was better compared to that following frontline surgery (HR: 0.72, 95% CI 0.69–0.76), in particular for the R0 resection rate (RR: 0.51; 95% CI 0.47–0.55). These conclusions should be interpreted with caution because of the heterogeneity of the studies pooled and their retrospective designs, as well as the lack of discrimination of subgroups for the administration or not of adjuvant chemotherapy [46].

##### Tumor Regression Score

The pathological response is a prognostic factor in operated patients. The robustness and reproducibility of validated classifications must be discussed. The most consensual classification at present is from the College of American Pathologists (CAP) based on the degree of radiation-induced fibrosis and regressive changes in the tumor (see Table 2) [47]. 

When assessing a patient’s response to induction therapy in a multidisciplinary team meeting, the most relevant criterion is CAP 0, which is found in no more than 5% of patients.

#### 4.2.2. Adjuvant Treatment According to Induction Chemotherapy Regimen?

##### Gemcitabine

The most robust data are available for gemcitabine, based on the results of PREOPANC-1, a phase III randomized trial with two arms: surgery plus adjuvant gemcitabine versus perioperative gemcitabine in borderline and resectable PDAC. It should be remembered that the results were positive for the primary endpoint of OS (HR 0.71; *p* = 0.047) supporting perioperative treatment, and the notion of perioperative treatment with the same chemotherapy before and after surgery [34,38] (see Table 3).

##### GemNab

The randomized phase II trial SWOG S1505 (NCT02562716) in patients with resectable PDAC evaluated a perioperative strategy using mFOLFIRINOX vs. GemNab, with 6 neoadjuvant cycles and 6 adjuvant cycles in case of surgery. In the preliminary results presented at the 2019 ASCO meeting, the resection rate in the two arms was 77% and 73%, respectively [48]. The updated results, presented at the virtual 2020 ASCO meeting, suggest that results are similar for efficacy and safety for the two chemotherapy combinations. The median OS (primary endpoint) was 22.4 months in the FOLFIRINOX arm versus 23.6 months in the GemNab arm (see Table 3).

In an intention to treat analysis, 71.5% of patients could undergo curative surgery and 60% receive adjuvant chemotherapy identical to the neoadjuvant chemotherapy.

OS data were not comparable to those reported in the APACT and PRODIGE 24 adjuvant trials in which patients were included postoperatively after a CT scan and CA 19-9 [49]. Data are pending for borderline or locally-advanced PDAC with the GemNab induction scheme.

##### FOLFIRINOX

There is significant heterogeneity in the postoperative adjuvant treatments proposed in the literature in patients who receive FOLFIRINOX induction treatment. In the first retrospective series [39], only half the patients (53.7%) received adjuvant chemotherapy, mainly gemcitabine (75% of cases) which was the standard before the publication of the ESPAC-4 trial (2017). Adjuvant chemotherapy did not influence DFS on univariate analysis (*p* = 0.620). The same group published a subsequent analysis including patients who had received preoperative CRT after FOLFIRINOX. Among them, 57.1% of patients had received adjuvant chemotherapy [36]. Once again, adjuvant chemotherapy did not have prognostic value for OS or PFS, whatever the treatment arm (FOLFIRINOX alone or FOLFIRINOX then CRT). Nevertheless, patients in the FOLFIRINOX group received adjuvant treatment more often than those in the FOLFIRINOX-CRT group (73.2% versus 41.2%; *p* = 0.002) and there was no specification of the type of adjuvant treatment. Besides their retrospective design, these studies were limited by the pooling of borderline and locally advanced PDAC and differences in the number of induction cycles. This may have influenced the decision to administer adjuvant chemotherapy and their modalities. 

International multicenter trials to define the optimal adjuvant chemotherapy, which is often similar to that used as induction treatment, are ongoing. For example, the PRODIGE48-PANACHE01 trial evaluated mFOLFIRINOX in patients who could receive frontline resection while the PRODIGE44-PANDAS trial is evaluating the value of a combination of CRT and induction FOLFIRINOX and proposes adjuvant monotherapy with modified LV5FU2 or gemcitabine.

One of the main limitations of perioperative regimens with FOLFIRINOX are neurotoxicity-induced sequelae, which prevent the administration of optimal doses of oxaliplatin in the adjuvant setting.

##### S1

In Japan, the phase II/III trial JSAP-05 randomized patients with resectable PDAC into two arms: a "peri-operative" arm with neoadjuvant chemotherapy consisting of 2 cycles of gemcitabine and S-1 followed by surgery, followed by an additional 4 cycles of adjuvant S-1, and an "adjuvant" arm which included frontline surgery followed by 4 cycles of adjuvant S-1. The median OS for the perioperative group was 36.7 versus 26.6 months in the adjuvant group, with an HR of 0.72 (95% CI 0.55–0.94; *p* = 0.015) with equivalent morbidity between the two groups. This trial thus suggests that the same chemotherapy should be continued post-operatively [50,51] (see Table 3).

### 4.3. What Is the Optimal Delay to Start Adjuvant Treatment?

Valle et al. [8] suggested that adjuvant chemotherapy should only be delayed until the patient has fully recovered from surgery, as long as the full protocol can be administered (i.e., 6-month schedule) [8]. This in depth analysis of the ESPAC-3 trial, which evaluated the best time to start chemotherapy after surgery and the best duration did not find any difference in survival between patients who started chemotherapy within 8 weeks after surgery and those who started after up to 12 weeks [8]. In another retrospective study including 488 patients results of delayed initiation of adjuvant chemotherapy (12 weeks after surgery) were the same as earlier administration, and both options were superior to observation [52]. Other retrospective studies have shown that the initiation date of adjuvant treatment after surgery does not influence overall survival. For example, the study by Turner et al. using registries from the National Cancer Database show that undergoing adjuvant chemotherapy is associated with improved overall survival in patients with stage I-III PDAC, even if it is delayed up to 24 weeks [53]. However, in another National Cancer Database study, survival in patients who began adjuvant therapy within 28 to 59 days after primary surgical resection was better than those who received adjuvant therapy before 28 days or after 59 days [54]. 

In meta-analysis by Petrelli et al., with 34 comparative studies assessing a total of 141,853 patients, the benefit of earlier chemotherapy (started between 6–8 weeks) was not evident in pancreatic cancer (HR = 1, 95% CI: 1–1.01; *p* = 0.37) in contrast to colon and gastric cancers [55].

The type and duration of adjuvant chemotherapy in patients who receive preoperative treatment is ill-defined. Theoretically, a 6-months sequence including neoadjuvant/induction and adjuvant chemotherapy is proposed. Persistent toxicity of neoadjuvant/induction chemotherapy (mainly neuropathy) probably influences this decision. Final results from trials evaluating peri-operative treatment such as FOLFIRINOX (PRODIGE48-PANACHE); GemNab (SWOG S1505, perioperative mFOLFIRINOX vs. GemNab in resectable PDAC) or Gemcitabine-S1 (JSAP-05) are awaited. A clinical trial precisely evaluating the type and duration of adjuvant strategy (<6 months or >6 months) could also be useful in the future in case of surgery after induction treatment.

### 4.4. In Which Patients?

According to international guidelines, adjuvant treatment may be administered in operated PDAC regardless of the stage of the tumor [4,5,7]. This recommendation will probably be revised in the future to include neoadjuvant/induction treatment.

In the recent ESPAC-5F trial, which evaluated neoadjuvant strategies in borderline PDAC patients, the rate of adjuvant chemotherapy was similar in the 4 arms, around 50%, with surgery alone (53%), GemCap (50%), mFOLFIRINOX (45%) and CRT (44%). In the ESPAC-5F trial, adjuvant chemotherapy was at the investigator’s discretion, depending on the guidelines chosen, but it was mainly 5FU- or gemcitabine-based for 6 months [35]. These results showed that 48.9% of the operated patients could receive adjuvant chemotherapy, which was confirmed in a large meta-analysis of 45 studies including 3359 patients (Araujo [56]). In that meta-analysis, patients who received less intensive adjuvant therapies were those who received induction therapy. There are two explanations for this: (1) the presence of treatment sequelae from induction therapy, mainly >grade 2 neuropathy related to oxaliplatin; (2) a severely impaired postoperative general status due to infectious complications, denutrition, fatigue or liver toxicity caused by preoperative chemotherapy, such as sinusoidal obstructive syndrome or severe steatosis due to oxaliplatin and irinotecan, respectively [27]. Intensive postoperative supportive care is absolutely needed to optimize recovery of patients before beginning adjuvant treatment. Another recent hypothesis is the role of the surgical procedure in the timing of the initiation of adjuvant chemotherapy. A recent study of 23,494 patients operated on for pancreatic cancer showed that compared to “low volume” hospitals patients in “high-case-volume” hospitals had the highest rates of adjuvant chemotherapy administration after pancreaticoduodenectomy and distal pancreatectomy. Moreover, compared to open surgery for all resection types, laparoscopic surgery was associated with a higher rate of adjuvant chemotherapy use at high and highest-case-volume hospitals and less delay in chemotherapy at high-volume hospitals [57]. 

Nevertheless, because of the reported excess mortality, it is impossible to recommend laparoscopy for pancreaticoduodenectomy based on current data, despite an equivalent quality of exeresis (R0, N analyzed). This excess mortality was found in the randomized LEOPARD-2 trial, resulting in the current guidelines which do not recommend the minimally invasive approach to pancreaticoduodenectomy (10 versus 2%; RR = 4.90; 95% CI: 0.59–40.44; *p* = 0.2) [58].

## 5. Paradigm Shift, Looking for Biomarkers of Adjuvant Chemotherapy Efficacy

Detection of infra-clinical circulating disease is a major challenge that could help optimize perioperative management of patients with resectable PDAC [59,60,61,62].

### 5.1. Biological Markers

#### 5.1.1. Circulating Tumor DNA

A recently published exploratory study in 112 patients with resectable PDAC has shown the prognostic value of circulating tumour DNA (ctDNA) [62]. PCR-based SafeSeqS assays were used to detect *KRAS* mutations at codons 12, 13, 61 and a statistical algorithm classified the ctDNA samples as detectable and non-detectable. Out of 42 available plasma samples, ctDNA *KRAS* mutations were detectable in 62% of cases pre-operatively and in 37% of cases post-operatively. After a median follow-up of 38.4 months, preoperative ctDNA detection was associated with significantly lower recurrence-free survival (HR 4.1; *p* = 0.002) and significantly lower OS (HR: 4.1; *p* = 0.015). Post-operatively detectable ctDNA was associated with significantly lower recurrence-free survival (HR 5.4; *p* < 0.0001) and OS (HR: 4.0; *p* = 0.003). Tumor recurrence occurred in 100% of patients with detectable ctDNA after surgery, including those who received adjuvant gemcitabine-based chemotherapy [62] (see Table 4). A meta-analysis of five studies pooling 375 patients also suggested that ctDNA was currently the most promising prognostic biomarker in resectable PDAC, either at baseline or postoperatively [61].

Before using biomarkers to routinely guide adjuvant chemotherapy, several issues must be resolved: (1) What technology should be used? At present ctDNA detection can also be performed by methylated markers, which is a widely available and less expensive technology than mutation detection [63]; (2) Should ctDNA be measured pre- or post-operatively? (3) When preoperative ctDNA is detected, is neoadjuvant treatment more suitable than frontline surgery? (4) Should secondary resection be limited to patients with undetectable ctDNA after neoadjuvant treatment? 

The intensity of adjuvant therapy and monitoring could be adjusted in the presence of post-operative detection of ctDNA. These points must be evaluated in prospective trials.

#### 5.1.2. CA19-9 Oncomarker

The CA 19-9 oncomarker is a classic prognostic marker of all stages of pancreatic cancer [4]. However, because the indications for adjuvant chemotherapy concern all stages of surgically treated pancreatic cancer [4,5,7], the use of this marker is a subject of debate, even though it is easy to measure in current practice. Indeed, international guidelines have added Ca 19-9 as a biological criterion for resectability [64]. 

Also, the phase III RTOG 9704 trial confirmed the prognostic value of CA19-9 as a post-operative marker in resected pancreatic cancers following post-operative CRT [65] (see Table 4). CA19-9 was found to have prognostic value in multivariate analysis whether it was measured as a continuous or categorical variable. In a recent retrospective study of 957 patients undergoing pancreactectomy for PDAC between 2000 and 2013, one of the major post-operative prognostic factors was CA19-9 > 37 U/mL (OR 3.38). However, none of the patients in that study had received neoadjuvant therapy [66]. The value of dosing CA19-9 and the best definition of cutoffs could be pertinent in a neoadjuvant/induction strategy to improve selection of the best candidates for surgery and to help determine the choice of adjuvant chemotherapy, if necessary. For example, in a recent single center retrospective study in patients with borderline or locally advanced pancreatic cancer who received induction treatment with FOLFIRINOX and underwent resection, preoperative CA 19-9 > 100 U/mL was predictive of shorter post-operative DFS and decreased OS [67]. 

Once again, clinical trials designed to improve adjuvant chemotherapy recommendations should take into account the CA19-9 marker as well as other emerging biomarkers such as ctDNA.

#### 5.1.3. Immune Inflammatory Markers

The predictive value of systemic inflammatory markers has been explored in various types of cancer to estimate cancer burden [68]. The preoperative neutrophil-to lymphocyte ratio (NLR) is considered to be a significant independent prognostic indicator in patients with resected PDAC [69]. Pretreatment NLR values were significantly associated with distant metastases in pancreatic cancer patients [70]. In a meta-analysis of retrospective studies with operated patients, a high pre-operative NLR indicates a worse prognosis than in patients with a low NLR (see Table 4). Unfortunately, the lack of consensus on an NLR cut-off value limits the use of these results in clinical practice [71]. 

Other inflammatory markers such as the lymphocyte-to-monocyte ratio could be useful. In a recent retrospective study, survival was significantly worse in patients with a low lymphocyte-to-monocyte ratio after neoadjuvant therapy (<3.0) than in those with a lymphocyte-to-monocyte ratio ≥3.0 (14.9 months vs. 31.7 months, *p* = 0.006). These results suggest that lymphocyte-to-monocyte ratios could play a potential role in the stratification of the treatment strategy in patients with borderline, resectable, pancreatic cancer [72]. 

### 5.2. Predictive Pathological Biomarkers

These biomarkers have been more extensively explored with gemcitabine [25]. The human equilibrative nucleoside transporter 1 (hENT1) is the main transporter responsible for the cellular absorption of gemcitabine. In the ancillary analysis of the ESPAC 3 trial, high expression of hENT1 on immunohistochemistry was reported to be a predictive marker of response to gemcitabine in the adjuvant setting [73,74] (see Table 4). However, discordant results have been reported in adjuvant and metastatic settings [25] as well as in relation to the antibody used (10D7G2 and SP120) for immunohistochemistry [74,75]. Phosphorylation by deoxycytidine kinase (dCK), which corresponds to the first step in the transformation of gemcitabine into an active metabolite, is another marker which has been described. Elevated dCK levels in immunohistochemistry have also been significantly associated with longer OS in patients treated with adjuvant gemcitabine [76]. However, generally, these markers cannot be used in clinical practice. Thus, before any clear recommendations can be made on the use of hENT1 as a predictive biomarker, standardized procedures to assess the expression of hENT1 and other biomarkers such as dCK should be established and validated in prospective trials.

### 5.3. Molecular Markers (see Table 4)

#### 5.3.1. Chemotherapy Signatures Using Organoids

To overcome the pauci-cellular state of primitive pancreatic tumors, organoid cultures could facilitate in-depth molecular characterization, through advances in high-throughput sequencing. Chemograms could be performed on these organoid cultures to guide treatment. This experimental approach requires evaluation in future clinical trials [77].

#### 5.3.2. Molecular Classifications

Molecular classifications were established with five subtypes of adenocarcinomas, based on transcriptomic profiles after bioinformatics analysis. The transcriptomic data were derived from high-throughput molecular screening of formalin-fixed paraffin-embedded tumor samples.

The subtypes “pure basal-like”, “stroma-activated”, “desmoplastic”, “immune classical” and “pure classical” have been identified to define the biological and micro-environmental diversity of pancreatic cancer [78]. These molecular classifications help identify patients who could respond to specific treatment with cytotoxic molecules or targeted therapy. However, their use in clinical practice is still limited and the methodology to assess these subtypes as well as the subtypes themselves are still a subject of debate. There is a consensus that there are two types of tumoral cells, basal-like and classical. The COMPASS study in metastatic patients receiving first-line FOLFIRINOX suggests that the subtype may have predictive value to help choose the best treatment, but a prospective study is needed to evaluate these results in the adjuvant or perioperative setting [79].

#### 5.3.3. Patients with Germinal or Somatic BRCA Mutation

The POLO Phase III trial in patients with metastatic PDAC and the germline *BRCA1/2* mutation (BRCAm) (5–7% of patients) has paved the way for targeted therapies. Maintenance therapy with the PARP inhibitor olaparib (300 mg twice daily) nearly doubled PFS from 7.4 months to 3.8 months; HR: 0.53; 95% CI 0.35–0.82; *p* = 0.004) in patients with controlled tumors after 16 weeks of platinum-based chemotherapy compared to placebo [80]. An interim analysis of OS with 46% mature data, showed no difference between the olaparib and placebo groups (median OS: 18.9 months vs. 18.1 months; HR for death: 0.91; 95% CI 0.56–1.46; *p* = 0.68). Quality of life was not impaired with olaparib [81]. It would be interesting to test the value of olaparib in patients with tumors containing somatic mutations as there seems to be a possible difference between somatic and germinal BRCA mutation status in relation to response ton parp-inhibitors [82]. In addition, treatment with PARP inhibitors for adjuvant therapy in BRCAm PDAC patients who have undergone surgical resection requires further study. 

#### 5.3.4. Patients with a High Microsatellite Instability/Mismatch Repair-Deficient Cancer

About 1% of PDAC patients present with high microsatellite instability (MSI). This tumor phenotype, which is often a feature of the Lynch syndrome, may be associated with a significant response to immune checkpoint inhibitors [83]. In contrast, the objective response rate of PDAC to these treatments was lower than that in other MSI cancers (18.2 % vs. 33–57.1% in other digestive, gynecological or brain tumors) as shown in the phase 2 KEYNOTE-158 trial [84]. 

## 6. Better Management of Adjuvant Setting with Supportive Care

One of the major challenges of adjuvant therapy in patients operated for PDAC is to improve the rate of patients who are eligible for chemotherapy. The more aggressive the perioperative treatment and the surgical procedures are, the more important supportive care becomes.

Otherwise, it has clearly been shown that performing surgery in high-volume, authorized centers with expert, multidisciplinary teams and intensive care units can help minimize operative morbidity and mortality [85]. 

While prehabilitation is important to limit the risk of postoperative complications [86], optimization of adjuvant therapy should be improved. Published data on when to initiate adjuvant chemotherapy are important for this purpose. 

The improved survival in recent trials has also been attributed to better management of supportive care by gastro/oncologists. Beside “classical” supportive actions (anxiety/depression, pain control, diarrhea, diabetes and nutritional), adapted physical activity (APA) could also improve both the quality of life and tolerance to chemotherapy but also reduce the risk of cancer relapse [87]. APA as a support option associated with chemotherapy is currently being evaluated in adjuvant clinical trials (for example, the PRODIGE56-APACAPop trial with quality of life as primary endpoint – NCT03400072)

Finally, the psychological dimension of this specific cancer should be taken into consideration in patients who are operated on and are considered to have a chance of long-term survival but also a theoretically high risk of tumor relapse. Improved characterization of anxiety and depressive disorders and their appropriate management are a challenge because of the important role they play in these patients [88].

## 7. Conclusions

The perioperative treatment of patients who have undergone tumor resection for PDAC has significantly progressed in the past two decades, especially since the use of modified FOLFIRINOX. There are currently several new challenges, in particular; (i) to better select patients for surgery by detecting metastases with modern imaging and new biomarkers such as circulating tumor DNA; (ii) to optimize the role as well as the timing, type and duration of neoadjuvant/induction and adjuvant therapies; and (iii) to promote patient quality of life by increasing multidisciplinary, supportive care to prevent or actively treat anxiety, denutrition, diarrhea and psychic deterioration.

## Figures and Tables

**Table 1 cancers-12-03866-t001:** Main Phase III studies evaluating adjuvant chemotherapy protocols.

Type of Chemotherapy	Design of the Study	DFS Results	OS Results	Reference
5FU	Phase 3, international, multicentric (*N* = 541). -CRT (20 Gy in 10 fractions/2 weeks with 500 mg/m^2^ 5FU IV on days 1–3, repeated after 2 weeks) -CT (IV 5FU 425 mg/m^2^ and folinic acid 20 mg/m^2^ daily for 5 days, monthly for 6 months).	CRT: -Median DFS: NA CT: -Median DFS: NA	CRT -Median OS: 15.5 months with CRT vs. 16.1 months without; HR: 1.18 (95% CI 0.90–1.55), *p* = 0.24 CT -Median OS: 19.7 months with CT vs. 14.0 months without; HR 0.66 (0.52–0.83), *p* = 0.0005.	ESPAC-1 [13]
Gemcitabine	Phase 3, international, multicentric (*N* = 368). -gemcitabine 1000 mg/m^2^ IV once a week for 3 of every 4 weeks for 6 months -or observation	Gemcitabine -Median DFS: 13.4 months (95% CI, 11.4–15.3) Observation -Median DFS 6.9 months (95% CI, 6.1–7.8); *p* < 0.001.	Gemcitabine -Median OS: 22.1 months (95% CI, 18.4–25.8) Observation -Median OS: 20.2 months (95% CI, 17–23.4)	ISRCTN34802808 [14]
Gemcitabine vs. 5FU	Phase 3, international, multicentric (*N* = 287) Chemotherapy -5FU IV 425 mg/m^2^ administered 1 to 5 days every 28 days or gemcitabine 1000 mg/m^2^ IV once a week for 3 of every 4 weeks for 6 months. -or observation	-Two chemotherapy groups Median DFS: 5FU: 23.0 months (95% CI, 17.0–51.9 months) Gemcitabine: 29.1 months (95% CI, 19.5–45.4 months) -Observation: 19.5 months (95% CI, 14.2–30.3 months)	-Two chemotherapy groups Median OS: 43.1 (95%, CI, 34.0–56.0); HR: 0.86 (95% CI, 0.66–1.1), *p* = 0.25. Observation: Median OS: 35.2 months (95% CI, 27.2–43.0)	ESPAC-3 [15]
GemCaP	Phase 3, open-label, international multicentric (*N* = 732) Randomisation 1:1 Six cycles of either 1000 mg/m^2^ gemcitabine alone administered once a week for three of every 4 weeks (one cycle) or with 1660 mg/m^2^ oral capecitabine (=GemCap group) administered for 21 days followed by 7 day’ rest (one cycle).	GemCap: -Median DFS 13.9 months (12.1–16.6) Gemcitabine -Median DFS 13.1 months (11.6–15.3); HR 0.86, 95% CI 0.73–1.02, *p* = 0.082.	GemCap Median OS: 28.0 months (95% CI 23.5–31.5) Gemcitabine Median OS: 25.5 months (22.7–27.9) HR: 0.82 (95% CI 0.68–0.98), *p* = 0.032.	ESPAC-4 [11]
GemNab	Phase 3, international, multicentric (*N* = 866) Randomisation 1:1 Nab-paclitaxel 125 mg/m^2^ + gemcitabine 1000 mg/m^2^ (GemNab group) or gemcitabine 1000 mg/m^2^ (Gem group) for 3 of every 4 weeks for 6 months.	GemNab: -Median independent reviewer-assessed DFS: 19.4 months Gemcitabine Median independent reviewer-assessed DFS: 18.8 months HR: 0.88 (95% CI, 0.729–1.063); *p* = 0.1824.	GemNab: -Interim OS: 40.5 months Gemcitabine: -Interim OS: 36.2 months HR, 0.82; 95% CI, 0.680–0.996; nominal *p* = 0.045).	APACT [10]
mFOLFIRINOX	Phase 3, international, multicentric (*N* = 493) mFOLFIRINOX regimen (combining 5FU 2400 mg/m^2^, irinotecan 150 mg/m^2^ and oxaliplatin 85 mg/m^2^ every 14 days for 12 cycles) versus gemcitabine 1000 mg/m^2^ during 6 months.	mFOLFIRINOX: -Median DFS: 21.6 months Gemcitabine: Median DFS: 12.8 months; stratified HR for cancer-related event, second cancer, or death: 0.58 (95% CI, 0.46–0.73); *p* < 0.001.	mFOLFIRINOX: -Median OS: 54.4 months (95% CI, 41.8 to not reached) Gemcitabine: -Median OS: 35.0 months (95% CI, 28.7 to 43.9) (stratified hazard ratio for death, 0.64; 95% CI, 0.48 to 0.86; *p* = 0.003)	PRODIGE 24 [9]
S1	Phase 3, multicentric, in Japan (*N* = 385) Randomised 1:1 gemcitabine 1000 mg/m^2^ IV once a week for 3 of every 4 weeks for 6 months or S-1 40 mg, 50 mg, or 60 mg according to body-surface area, orally administered twice a day for 28 days followed by a 14 days rest, every 6 weeks (one cycle), for up to 4 cycles.	S1: Median DFS 22.9 months (17.4–30.6) Gemcitabine: Median DFS: 11.3 months (95% CI 9.7–13.6) -HR for relapse 0.60 (95% CI 0.47–0.76, *p* < 0·0001).	S1: Median OS: 46.5 months (37.8–63.7) Gemcitabine: Median OS: 25.5 months (95% CI 22.5–29.6) -HR of mortality: 0.57 (95% CI 0.44–0.72), *p* non-inferiority < 0.0001, *p* < 0.0001 for superiority	JASPAC-01 [12]

CRT: chemoradiotherapy, CT: chemotherapy, DFS: disease-free survival, NA: not applicable, HR: hazard ratio, OS: overall survival.

**Table 2 cancers-12-03866-t002:** Classification of the College of American Pathologists for Treated Pancreatic Ductal Adenocarcinoma (Ryan, Histopathology, 2005).

CAP Score Index	Description
0	No viable cancer cells (complete response)
1	Single cells or rare small groups of cancer cells (near complete response)
2	Residual cancer with evident tumor regression,
3	Extensive residual cancer with no evident tumor regression (poor or no response)

**Table 3 cancers-12-03866-t003:** Main Phase II/III studies evaluating peri-operative chemotherapy protocols.

Type of Chemotherapy	Design of the Study	DFS Results	OS Results	Reference
Gemcitabine	Randomized phase III multicentric (*N* = 248) Patients with (borderline) resectable pancreatic cancer Arm A: immediate surgery Arm B: preoperative CRT Both followed by adjuvant gemcitabine. The preoperative CRT consisted of 15 times of 2.4 Gy combined with gemcitabine 1000 mg/m^2^ on days 1, 8 and 15, preceded and followed by a cycle of gemcitabine.	Median DFS Arm A: 7.7 months Arm B: 8.1 months; HR 0.73; *p* = 0.032.	Median OS Arm A: 13.5 months Arm B: 17.1 months; HR 0.71; *p* = 0.047.	PREOPANC-1 [34,38]
Gemcitabine vs. mFOLFIRINOX	Randomized phase II multicentric trial (*N* = 102) Patients with resectable pancreatic cancer. Peri-op CT (12 weeks pre-, 12 weeks post-op) with either mFOLFIRINOX (Arm 1) or Gem/nab (Arm 2).	Median DFS: Arm 1: 10.9 months Arm 2: 14.2 months *p* = 0.87	Median OS: Arm 1: 22.4 months Arm 2: 23.6 months	SWOG S1505 [49]
S1	Randomized phase II/III multicentric trial (*N* = 364) Resectable PDAC Neoadjuvant chemotherapy with gemcitabine + S1 or upfront surgery. Adjuvant S-1 was administered for 6 months to patients with curative resection.	Median DFS: not yet communicated	Median OS Neoadjuvant Gemcitabine + S1: 36.7 month Upfront surgery+adjuvant S1: 26.6 months; HR 0.72 (95% CI 0.55–0.94); *p* = 0.015.	JSAP-05 [50,51]

CRT: chemoradiotherapy, CT: chemotherapy, Gy: Gray, OS: overall survival.

**Table 4 cancers-12-03866-t004:** Promising biomarkers of adjuvant chemotherapy after pancreatic surgery, evaluated in prospective studies.

Prognostic Factor	Design, Type of Study	*N*	Multivariate Analysis			Reference
			HR (95% CI)	Outcome	*p*-Value	
Biological markers						
Pre-operative ctDNA	Pre- and post-operative samples for ctDNA analysis collected. PCR-based-SafeSeqS assays used to identify mutations of KRAS in the primary tumor and to detect ctDNA	112 patients in the study In 42 plasma available samples, KRAS mutated ctDNA detected in 62% (23/37) pre-operative and 37% (13/35) post-operative	HR 4.1	RFS	*p* = 0.002	[62]
			HR: 4.1	OS	*p* = 0.015	
Post-operative ctDNA			HR: 5.4	RFS	*p* < 0.0001	
			HR: 4.0	OS	*p* = 0.003	
CA 19-9	Prospective analysis of CA 19-9 levels in patients treated with adjuvant CRT in RTOG 9704 phase III trial	385 patients with assessable CA 19-9	HR: 3.53	Cutoff 180 significant survival difference favoring patients with CA 19-9 < 180	*p* < 0.0001	[65]
			HR: 3.4	Cutoff 90 significant survival difference in patients with CA 19-9 < or = 90 (HR, 3.4; *p* < 0.0001)	*p* < 0.0001	
Neutrophil to lymphocyte ratio	Meta-analysis retrospective studies	1519 patients (8 studies)	Pooled HR: 1.77	Association between a “high” pre-operative NLR and OS.	*p* < 0.01	[71]
**Pathological markers**						
hENT1	Ancillary study of ESPAC3 Microarrays from 434 patients randomized to chemotherapy in the ESPAC-3 trial (plus controls from ESPAC-1/3) were stained with the 10D7G2 anti-hENT1 antibody	380 patients (87.6%) and 1808 cores were suitable and included in the final analysis.	HR: 9.87	“Low” hENT1 expression: Median survival with gemcitabine 17.1 (95% CI = 14.3 to 23.8) months	*p* = 0.002	[73]
				“high” hENT1 expression: Median survival: 26.2 (95% CI = 21.2 to 31.4) months		
**Molecular markers**						
BRCA	Prospective studies are warranted in adjuvant setting					[80]
MSI-H	Prospective studies are warranted in adjuvant setting					[84]
Organoids	Prospective studies are warranted in adjuvant setting					[77]
Molecular classifications	Prospective studies are warranted in adjuvant setting					[78]

ctDNA: circulating tumour DNA, HR: Hazard Ratio, MSI-H: High Microsatellite Instability, OS: Overall Survival, PCR: Polymerase Chain Reaction, RFS: Recurrence-Free Survival.

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
