# Peer review of "Adjuvant Pancreatic Cancer Management: Towards New Perspectives in 2021"

_cancers, 2020, doi:10.3390/cancers12123866_

Round 1

Reviewer 1 Report

In this manuscript, Turpin et al. attempted to summarize evidence-based adjuvant management of resected pancreatic cancer, and to raise current and future concerns, especially the need for biomarkers and the best holistic care for patients. The topic is interesting. This manuscript is well organized and comprehensively described.

 But there were too many typos and further careful examination with correction should be done before acceptance.

  1. In the References, items 4 and 19 are the same (duplicated). So are items 9 and 16, 61 and 62.
  2. In Line 19 of Page 3, “a mortality of HR” should be corrected as “a HR of mortality”.
  3. In the 3.1 paragraph of page 5, “…with OS graphs only beginning late after 2 years“ should be “…with OS curve separation only beginning late after 2 years“, and “…over gemcitabine alone at a dose of >92.5 IU/mL” may be modified as “…over gemcitabine alone at a CA19-9 level of >92.5 IU/mL”.
  4. In the 4.1 paragraph of Page 6, the abbreviation “LAPC” should be spelled out. The “LAPACT phase II open-label trial” should be cited with a reference. And the listed references “(Pietrasz 2019, Murphy 2019, Thienhoven JCO, 2018)” should be “(Pietrasz 2019, Murphy 2019, Van Tienhoven 2018)”
  5. In the 4.2.1 Tumor regression score paragraph of Page 8, “American College of Pathologists (CAP)” should be “College of American Pathologists (CAP)”.
  6. In the 4.2.2 GemNab paragraph of Page 9, ”(Sohal D vASCO2020)” should be “(Sohal D 2020)”.
  7. In the 4.4 paragraph of Page 11, “(Ghaneh P et al., abstr. 4505)”should be “(Ghaneh P 2020)”.
  8. In the 5.1.1 paragraph of Page 12, “cDNA samples as detectable and non-detectable” should be “ctDNA samples as detectable and non-detectable”; and “preoperative cDNA detection was associated with significantly lower recurrence-free survival” should be “preoperative ctDNA detection was associated with significantly lower recurrence-free survival”.

Author Response

Dear Reviewer 1,

Thank you for your constructive comments.

Reviewer 2 Report

The manuscript is well-written, although there are several sentences are not clear and its create confusing. Therefore, it is recommended that this contribution be accepted after minor revision.

Reviewer 3 Report

This is a well-written,updated, and comprehensive review on this evolving topic.

Minor point:

2.1 when citing ESPAC-1 study please report 5-FU modulated by leucovorin.

Dealing with Prodige 24,inclusion criteria should not be viewed as criticisms(pre chemo CT scan,Ca19.9 <180,good PS)but as the best pts' selection for a modern adjuvant trial instead.

3.1 GEMCAP methodological limitations

Last line: "at a level of..." instead at  a dose of

4.1 Induction strategy

the results of the GAP study should be included as the only randomized study

confronting GEM vs Abra/gem in this setting

ESMO 2019,Annals of oncology

673PD

Nab-paclitaxel (Nab) plus gemcitabine (G) is more effective than G alone in locally advanced, unresectable pancreatic cancer (LAUPC): The GAP trial, a GISCAD phase II comparative randomized trial

S. Cascinu1, R. Berardi2, R. Bianco3, D. Bilancia4, A. Zaniboni5, D. Ferrari6, S. Mosconi7, A. Spallanzani8, L. Cavanna9, S. Leo10, F. Negri11, G.D. Beretta12, A. Sobrero13,
M. Banzi14, A. Morabito15, A. Bittoni2, R. Marciano3, D. Ferrara4, S. Noventa5,
M.C. Piccirillo16

4.3  What is the optimal delay

The paper and metanalysis by Petrelli et al should be added

  Timing of Adjuvant Chemotherapy and Survival in Colorectal, Gastric, and Pancreatic Cancer. A Systematic Review and Meta-Analysis. Petrelli F, Zaniboni A, Ghidini A, Ghidini M, Turati L, Pizzo C, Ratti M, Libertini M, Tomasello G.Cancers (Basel). 2019 Apr 17;11(4):550. doi: 10.3390/cancers11040550.    
